# Stabilization of Silver Nanoparticles by Cationic Aminoethyl Methacrylate Copolymers in Aqueous Media—Effects of Component Ratios and Molar Masses of Copolymers

**DOI:** 10.3390/polym11101647

**Published:** 2019-10-10

**Authors:** Mariya E. Mikhailova, Anna S. Senchukova, Alexey A. Lezov, Alexander S. Gubarev, Anne -K. Trützschler, Ulrich S. Schubert, Nikolay V. Tsvetkov

**Affiliations:** 1Department of Molecular Biophysics and Polymer Physics, St. Petersburg State University, 199034 St. Petersburg, Russia; m.e.mikhailova@spbu.ru (M.E.M.); grease_91@mail.ru (A.S.S.); a.a.lezov@spbu.ru (A.A.L.); a.gubarev@spbu.ru (A.S.G.); 2Laboratory of Organic and Macromolecular Chemistry (IOMC), Friedrich Schiller University Jena, Humboldtstr. 10, 07743 Jena, Germany; anne-kristin.truetzschler@uni-jena.de (A.-K.T.); Ulrich.Schubert@uni-jena.de (U.S.S.); 3Jena Center for Soft Matter (JCSM), Friedrich Schiller University Jena, Philosophenweg 7, 07743 Jena, Germany

**Keywords:** Nanoparticles, Synthetic design of polymer structures in solution, Analytical ultracentrifugation, Light scattering, Colloid stability

## Abstract

The ability of aminoethyl methacrylate cationic copolymers to stabilize silver nanoparticles in water was investigated. Sodium borohydride (NaBH_4_) was employed as a reducing agent for the preparation of silver nanoparticles. The objects were studied by ultraviolet-visible (UV-vis) spectroscopy, dynamic light scattering (DLS), analytical ultracentrifugation (AUC) and scanning electron microscopy (SEM). Formation of nanoparticles in different conditions was investigated by varying ratios between components (silver salt, reducing agent and polymer) and molar masses of copolymers. As a result, we were successful in obtaining nanoparticles with a relatively narrow size distribution that were stable for more than six months. Consistent information on nanoparticle size was obtained. The holding capacity of the copolymer was studied.

## 1. Introduction

In recent decades, there has been increasing interest in synthesis and studies of nanostructures of various natures and the materials based on these nanostructures, including metal-polymer nanocomposites [1,2,3]. This interest in metal nanoparticles (NPs) is mainly related to their unique characteristics that differ significantly from the properties of their “large” analogues. Considerable attention is being focused on strategies for the preparation of monodisperse colloid solutions of noble metal nanoparticles, since they possess pronounced antibacterial properties [2,4,5]. Moreover, they are promising source materials for making biosensors, photonic crystals, porous membranes, microlenses, for applications in colloidal lithography and so forth [6,7,8,9,10]. Silver nanoparticles became the most popular material due to their relatively low cost and numerous possible applications in biotechnology; in particular, they are widely used in diagnostics and the treatment of oncological diseases [5,11]. 

However, colloidal solutions of silver tend to oxidize and aggregate, which results in their instability in aqueous media. One method for stabilizing silver nanoparticles in solution is the use of polymers of various architectures (linear, comb-shaped and branched). Moreover, reduction of silver ions in the presence of polymers makes it possible to control composition and size of nanoparticles, their size distribution and shape [12,13]. For this purpose, both synthetic (polyvinylpyrrolidone and poly(vinyl alcohol) [14,15]) and natural polymers (e.g., polysaccharides) [16,17,18] are employed. 

Polyelectrolytes have a highly important place among the polymers used for stabilizing dispersions of metal NPs [1,19,20,21]. In addition, these polymers may demonstrate antimicrobial, antitumor, anti-inflammatory and antioxidant properties and, thus, enhance the corresponding characteristics of silver NPs [5,22]. Of special note are polycations containing amino groups; these macromolecules have become deservedly popular due to their rather easy synthesis, variability of properties, promising applications in various fields. Such polymers are currently used as gene carriers (in design of gene vectors), in therapy of various diseases [23]. It is possible to extend the areas of application of cationic polymers and finely to tune their characteristics for performing a given task by copolymerization of monomers with various chemical structures [24].

Researchers engaged in studies and interpretation of properties of complex supramolecular structures (such as polymer/NP complexes) face many difficulties and among them is considerable polydispersity of the products. This polydispersity is caused, among other factors, by the broad molar mass distributions of the initial polymeric components. It is possible to reduce the dispersity of stabilizing polymers using controlled radical polymerization such as the reversible addition-fragmentation chain transfer (RAFT) process. This approach enables the synthesis of polymers with sufficiently narrow molar mass distributions [24,25,26]. 

Recently, we have demonstrated possibility of stabilization of silver nanoparticles by the cationic copolymer based on poly(aminoethyl methacrylate) [27]. It was shown that the nanoparticles synthesized by chemical reduction in the presence of this copolymer demonstrate satisfactory spectral characteristics in solution; besides, they are stable in solution for prolonged periods of time (more than six months).

The goal of the present work was to investigate the capability of cationic copolymer poly((2-aminoethyl)methacrylate-*co*-*N*-methyl(2-aminoethyl)methacrylate-*co*-*N*,*N*-dimethyl(2-aminoethyl)meth¬acrylate), synthesized by RAFT polymerization [25], for stabilization of dispersions of silver NPs in an aqueous medium. Besides, we intended to achieve a decrease in size polydispersity of the samples and to improve the spectral characteristics of the stabilized nanoparticles. We also analysed the influence of molar mass of the copolymer on its stabilizing properties and studied hydrodynamic characteristics of the resulting silver nanoparticles stabilized by the cationic macromolecules. 

## 2. Materials, Methods and Synthetic Procedure for Stabilized Nanoparticles

### 2.1. Methods

Ultraviolet-visible (UV-vis) spectroscopy was used as the main non-invasive method for detecting appearance of nanoparticles and monitoring dynamics of their formation and stabilization (UV-1800 spectrophotometer, Shimadzu Corp., Kyoto, Japan). The experiments were performed with a resolution of 1 nm, in the wavelength range λ∈ [190 to 1100] nm; the samples were placed in the quartz cell with an optical path length of 0.5 cm. 

Dynamic light scattering (DLS) experiments were carried out with the aid of a “PhotoCor Complex” apparatus (Photocor Instruments, Inc., Moscow, Russia). Two single-mode solid-state linear polarized lasers (λ_0_ = 654 nm) served as excitation sources. The experiments were performed at scattering angles (θ) ranging from 30° to 130°. The normalized intensity homodyne autocorrelation functions were fitted using the inverse Laplace transform (ILT) regularization procedure incorporated in “DynaLS” program (which provides distributions of relaxation times and hydrodynamic radii) and cumulant analysis (which provides the mean values of hydrodynamic radius <R_h_>_c_ and polydispersity index (PDI), www.photocor.ru/dynals/) [28,29]. Since all the observed modes (i) displayed diffusion nature (1/τ = D_i_q^2^), the values of translational diffusion coefficients D_i_ were calculated from the slope of linear dependence of the inverse relaxation time 1/τ on scattering vector squared q2=(4πn0λ0)2sin(θ2)2. The D_0i_ values were calculated from linear extrapolations of the D_i_(c) dependences to infinite dilution. Hydrodynamic radii of nanoparticles (R_hi_) were calculated using the Stokes-Einstein equation R_hi_ = kT/6πη_0_D_0i_, where k is the Boltzmann constant (1.38 × 10^−^^16^ erg K^−^^1^), T is the absolute temperature. Average values of hydrodynamic radii <R_h_>_c_ and PDI were calculated based on the cumulant analysis for scattering angle 90°.

The sedimentation velocity analytical ultracentrifugation (AUC) experiments were performed using a Beckman XLI analytical ultracentrifuge (ProteomeLab XLI Protein Characterization System) at rotor speeds of 3000, 5000 and 42,000 rpm depending on a sample. In the majority of experiments, aluminium centrepieces with an optical path length of 12 mm were used. The sample and reference sectors were loaded with 0.42 mL of a studied solution and solvent, respectively. The centrifuge camera was vacuumized and thermostabilized at 25 °C for at least one hour before the run. Both optical systems (interference and absorbance) were engaged in the experiments. The sedimentation velocity data were analysed using the Sedfit program [30]. Sedfit allows the obtaining of the distribution of sedimentation coefficients using the Provencher regularization procedure [31]. Two major characteristics of sedimentation velocity experiments were determined, viz. the sedimentation coefficient s and the frictional ratio (*f/f_sph_*). Both characteristics (s and *f/f_sph_*) should be extrapolated to zero solute concentration. Since the hydrodynamic investigations are usually performed in extremely dilute solutions, the linear approximations $s^{-1}=\left(s_{0}^{-1}\right)\left(1+k_{s}c+\ldots \right)$ and $f/f_{sph}={\left(f/f_{sph}\right)}_{0}\left(1+k_{f}c+\ldots \right)$ (where *k_s_* is the Gralen coefficient, *c* is the solution concentration, *f* is the translational friction coefficient, *f_sph_* is the translational friction coefficient of equivalent sphere) can be used for the extrapolations. In this manner, it is possible to determine hydrodynamic parameters $s_{0}$ and ${\left(f/f_{sph}\right)}_{0}$, which characterize a macromolecule at the infinite dilution limit. In some cases, the diffusion coefficient $D_{0sf}$ may be estimated from the frictional ratio calculated by Sedfit program: $D_{0sf}=\frac{k_{B}T{\left(1-\bar{\upsilon }\rho_{0}\right)}^{1/2}}{\eta_{0}^{3/2}9\pi \sqrt{2}{\left({\left(f/f_{sph}\right)}_{0}\right)}^{3/2}{\left(s_{0}\bar{\upsilon }\right)}^{1/2}}$ (where $k_{B}$ is the Boltzmann constant, T is the absolute temperature, $\bar{\upsilon }$ is the partial specific volume, $\rho_{0}$, $\eta_{0}$ are the solvent density and viscosity, respectively). Together with $D_{0sf}$, the value of hydrodynamic radius $R_{sf}$ may be estimated using the Stokes-Einstein relationship. To eliminate the common solvent properties, the intrinsic values of sedimentation coefficient [s] may be used: $\left[s\right]=s_{0}\eta_{0}/\left(1-\bar{\upsilon }\rho_{0}\right)$

Microphotographs of stabilized nanoparticles were obtained by scanning electron microscopy (SEM) (Zeiss Merlin, Carl Zeiss SMT, Oberkochen, Germany) on different scales (indicated in the images). Accelerating voltage (U_eht_) was 21.00 kV; chamber pressure was 50 to 70 Pa; working distance (WD) = 10 to 12 mm. The SE2 detector in high resolution column mode was used. The samples were obtained by drying drops of solutions on silicon wafers at 45 °C. The data have been processed with the free and open Source software Gwyddion (http://gwyddion.net/). Size distributions were obtained by finding the symmetric radii from over 100 images of individual NPs on SEM images in each case and directly counting the number of particles falling in different intervals (with an increment of 1 nm).

Viscometry and densitometry were used as auxiliary methods. Intrinsic viscosity was calculated from the data of measurements taken using a Lovis 2000 M microviscometer (Anton Paar, Graz, Austria). Density measurements were performed with a DMA 5000 M density meter (Anton Paar, Graz, Austria).

### 2.2. Materials

A comprehensive study of ternary copolymer poly((2-aminoethyl)methacrylate-*co*-*N*-methyl(2-aminoethyl)methacrylate-*co*-*N*,*N*-dimethyl(2-aminoethyl)methacrylate) over a wide range of molar masses (M_polymer_) has been recently performed using hydrodynamic and optical methods (Figure 1) [25].

Silver nanoparticles (NPs) were prepared by reduction of AgNO_3_ with NaBH_4_ in the presence of ternary cationic copolymer in water. Before the synthesis, chloride ions were substituted with nitrate ions by dialysis against concentrated solution of sodium nitrate followed by dialysis against distilled water. Copolymer concentrations in solutions after dialysis were determined from the density increment of aqueous solution (dp/dc = 0.250) for the copolymer [25]. The molar masses of polymers after dialysis were confirmed by measurements of intrinsic viscosity and diffusion coefficient (by DLS) in 0.2 M NaNO_3_ solution using the value of hydrodynamic invariant (A_0_) obtained earlier [25].

The following low molar mass reagents were obtained from Vecton (Russia)—Silver nitrate (AgNO_3_), sodium borohydride (the reducing agent) and sodium nitrate NaNO_3_. 

### 2.3. Synthetic Procedure for Nanoparticles

With the purpose of monitoring the effect of the reducing agent fraction on the process of NPs formation, two concentrations were held constant in all final mixtures—concentration of monomer units in solution c_mu_ = 4.8 ± 0.2 mM and concentration of silver nitrate c_AgNO3_ = 5.49 ± 0.05 mM. Thus, the molar ratio between AgNO_3_ and monomer units was 1.14 ± 0.05. The NaBH_4_:AgNO_3_ molar ratio was varied within the following limits: 1.85; 0.99; 0.72; 0.49; 0.23 (Δ(NaBH_4_:AgNO_3_) ≤ 0.05). The studies were carried out using the copolymer with molar mass M_polymer_ = 27.6 kDa.

Holding capacity of the copolymer macromolecules was tested by varying the molar ratio between AgNO_3_ and monomer units; the following values were used: 1.11 ± 0.05; 2.45 ± 0.05; 5.00 ± 0.05; 10.20 ± 0.05. The reducing agent was introduced at continuous stirring; the NaBH_4_:AgNO_3_ molar ratio was 0.96 ± 0.03 in all cases. The concentration of monomer units c_mu_ = 4.9 ± 0.2 mM in the final mixtures remained constant. The studies were carried out using the copolymer with molar mass M_polymer_ = 27.6 kDa

For monitoring the effect of molar mass on the formation and stabilization of nanoparticles by copolymers, the samples were prepared by mixing aqueous solutions of AgNO_3_ and ternary copolymers of various molar masses (M_polymer_ = 16.1, 27.6, 72.2 and 129.4 kDa) followed by introducing freshly prepared NaBH_4_ solution at continuous stirring. The concentrations and molar ratios in the final mixtures were as follows: c_mu_ = 4.6 ± 0.3 mM, c_AgNO3_ = 22.4 ± 1.6 mM, c_NaBH4_ = 21.4 ± 1.5 mM, (AgNO_3_:monomer units) = 4.88 ± 0.16 and (NaBH_4_:AgNO_3_) = 0.96 ± 0.01. 

## 3. Results and Discussion

### 3.1. The Effect of the Fraction of Reducing Agent

As noted above, recently we have demonstrated the possibility of successful stabilization of silver nanoparticles in aqueous solutions in the presence of a cationic copolymer by transformation of Ag(+) into Ag(0) with the use of NaBH_4_ as a reducing agent [27]. 

In the present work, we studied the influence of relative fraction of the reducer on the process of stabilization and final size distribution of NPs. To this end, we used constant concentrations of AgNO_3_ and monomer units in the final mixtures while varying the NaBH_4_:AgNO_3_ molar ratio in the 1.85 to 0.23 range (Table 1).

Figure 2a presents the absorption spectra taken after establishing equilibrium in the system, the equilibrium was confirmed in six days after introduction of the reducing agent. It should be noted that absorption spectra did not change for at least three months and further (Appendix A). In order to obtain the presented absorption spectra, the final mixtures were diluted tenfold. The positions of absorption peaks λmax and the values of full width at half maximum (fwhm), which characterizes the width of a surface plasmon resonance (SPR) band, are presented in Table 1.

Analysis of the presented results showed that an increase in the fraction of reducing agent leads to an increase in NP concentration (which is proportional to the absorption maximum intensity) [18,32,33]. The dependence of intensity of maximum in absorption spectrum on reducer fraction in the equilibrium mixtures (Figure 2b) is linear in the [0.23 to 0.99] range of molar ratios.

In this range, we also observed decrease in polydispersity of the obtained nanoparticles (i.e., decrease in the fwhm value, Table 1). Further increase in reducer fraction results only in an increase in the fwhm parameter. The position of SPR peak, which is almost similar for all presented cases (λ_max_, Table 1), indicates that silver nanoparticles stabilized by polymer chains are also approximately similar in size. Prolonged (three months and more) stability of absorption spectra confirmed our earlier conclusion: Cationic copolymer poly(AEMA-co-MAEMA-co-DMAEMA) does not participate in the reduction of Ag(+) and only stabilizes Ag nanoparticles [27]. This conclusion is also true when the solution contains stabilized nanoparticles and non-reduced silver ions Ag(+) (for molar ratios (NaBH_4_:AgNO_3_) < 1).

The stable systems containing nanoparticles were studied by dynamic light scattering and scanning electron microscopy.

The SEM image (Figure 2c and Appendix A) and distributions of symmetrized radii observed in microphotographs of nanoparticles (Figure 2d) are in good agreement with the UV-vis spectroscopy data. Table 1 provides the positions of maximums of distributions (R_SEM_) and full widths at half maximum of these distributions X (in parentheses after the R_SEM_ value). It should be noted that increase in the relative value of the reducer resulted in narrower size distributions of the final NPs.

Dynamic light scattering provided an opportunity to obtain information about the size of the nanoparticles stabilized by polymer chains in solutions. Note that SEM allows one to obtain images of metal NPs on a substrate surrounded by a dried polymer and cannot provide detailed information on the properties of the studied system in solutions. Figure 3 presents normalized distribution of hydrodynamic radii obtained by regularization procedure from autocorrelation function of light scattering intensity for samples with various molar ratios between NaBH_4_ and AgNO_3_.

Diffusion coefficients and hydrodynamic radii of the objects present in mixtures were estimated from the data of dynamic light scattering (Appendix A, Table 1). The first observed radius R_h1_ exceeds the size of individual polymer molecules (3.5 nm) [25] and can be related both to individual macromolecules and to individual nanoparticles stabilized by one or several polymer molecules. The second observed radius corresponds to aggregates of stabilized NPs. 

The estimated hydrodynamic radii are somewhat higher than those obtained by SEM, which illustrates different possibilities of the methods.

Average values of hydrodynamic radii <R_h_>_c_ and PDI (Table 1) based on the cumulant analysis demonstrated that the studied systems demonstrate size dispersity. The dependence of <R_h_>_c_ on molar ratios (NaBH_4_:AgNO_3_) was not observed in the studied range. However, for examples, for particles with optically equivalent characteristics, the contribution of each component to the total light scattering intensity is proportional to the volume and for particles with a shape close to their spherical radius in the cube I ~ V ~ R^3^ [29]. Thus, the observed values are strongly influenced by the presence of even a small proportion of aggregates in the studied system. 

Thus, the system with the lowest dispersity (according to the fwhm) containing stabilized nanoparticles (and the highest concentration of the final NPs) was prepared at the NaBH_4_:AgNO_3_ molar ratio close to 1. This ratio was selected for the investigations described in the following section.

### 3.2. Holding Capacity of the Copolymers

Holding capacity of the copolymers was studied using the mixtures with constant concentrations of monomer units and molar ratio (NaBH_4_:AgNO_3_) = 0.96 ± 0.03, while varying molar ratio (AgNO_3_:monomer units) in the range from 1.11 to 10.2.

Figure 4a presents the absorption spectra of stable mixtures (taken two months after introduction of the reducing agent) normalized to the same concentration (c_mu_ = 0.5 mM) of stabilizing polymer in solution (assuming that the Beer-Lambert-Bouguer law is valid). The enhanced intensity of SPR band is indicative of the following: Rise in the AgNO_3_ fraction in the mixtures before introducing NaBH_4_ led to a significant increase in concentration of the stabilized nanoparticles in the final mixtures (Appendix A). After completion of stabilization process (which proceeded for a week, similarly to the previous case), absorption spectra of the mixtures did not change for at least three months and further. For spectrophotometric studies, the mixtures were diluted from 10- and 100-fold for the samples range with (AgNO_3_:monomer units) ratios equal from 1.11 to 10.2, respectively.

It should be noted that concentration of stabilized nanoparticles in solution increases linearly with increasing molar ratios between AgNO_3_ and monomer units in the [1.1–5] range. Only when this ratio reaches 10.2, the dependence of absorption peak intensity on relative fraction of silver in the system deviates from linearity (Figure 4b). Besides, a shoulder appears in the long-wave region of the absorption spectrum (Figure 4a) that indicates increase in concentration of aggregates in the system. The fwhm value falls in the range of (AgNO_3_:monomer units) molar ratios between 1.1 and 5 (Table 2). Further increase in silver amount leads to increase in polydispersity of stabilized nanoparticles in the system.

Apparently, these observed features of absorptions spectra is due to the fact that after the introduction of the reducing agent the recovery and stabilization processes occur simultaneously. With a relatively small amount of recoverable silver, the number of monomer units of the polymer is sufficient to effectively stabilize arising and prevent the growth of large silver NPs. However, with further increase in silver content, the dependence of holding capacity on molar ratio (AgNO_3_:monomer units) deviates from linearity and polydispersity of the final NPs increases. The appearance of a number of large NPs (detected in the broadening of the absorption spectra and SEM distributions) leads to a decrease in the total surface of NPs present in the solution and thus eliminates the lack of polymer necessary for stabilization.

The NP-containing systems stable for three and more months were also studied by dynamic light scattering and scanning electron microscopy.

The DLS results (regularization procedure) demonstrate reduction in sizes of both individual NPs and aggregates in solutions with increasing silver fraction; further, the size of individual stabilized nanoparticles increases in the case when AgNO_3_:monomer units = 10.2 (Table 2). The values of <R_h_>_c_ based on the cumulant analysis (Table 2) are in good agreement with regularization procedure analysis. PDI’s are constant in whole range of molar ratios (AgNO_3_:monomer units) and mean values of <R_h_>_c_ decrease with rising of the molar ratio (AgNO_3_:monomer units) from 1.11 to 5.00. At 10.2 molar ratio regularization procedure demonstrates mono peak distribution. However, the PDI at 10.2 molar ratio is rather high. Apparently, in this cases the R_h1_/R_h2_ ratio is lower than 2, which coincide with maximum resolution of regularization procedure in DLS.

The results of SEM studies agree well with this observation: The size distributions for the cases of various molar ratios (AgNO_3_:monomer units) (Figure 4c) demonstrate a decrease in size polydispersity of NPs with increasing silver fraction followed by increase in polydispersity for the case (AgNO_3_:monomer units) = 10.2 (Appendix A). Figure 4d presents the microphotograph obtained for the sample with (AgNO_3_:monomer units) = 5.00.

The obtained results are interesting primarily from a practical point of view, since they indicate a molar ratios (AgNO_3_:monomer units) range in which a significant increase in the number of finely dispersed silver NPs with a minimum amount of stabilizing polymer can be achieved.

Thus, the results described in two previous sections of this contribution allowed us to make the following conclusion: When the molar ratio between NaBH_4_ and AgNO_3_ is maintained constant (close to 1) and the (AgNO_3_:monomer units) ratio is close to 5.00, we obtain the system containing nanoparticles with the narrowest size distribution; besides, the highest concentration of nanoparticles in the mixture is observed.

It is these parameters that were fixed at the following stage of our research (the study of influence of molecular mass of the stabilizing polymer on size distribution of the obtained nanoparticles, their stability and optical characteristics). Concentration of monomer units in the final mixtures was also held constant (c_mu_ = 4.6 ± 0.3 mM). These experimental conditions allowed us to eliminate the influence of concentration factors and make conclusions about the influence of polymer chain length on its ability to stabilize nanoparticles.

### 3.3. The Effect of Molar Mass

Use of the homologous series of copolymers with narrow molar mass distributions in the wide range of molar masses allowed us to obtain information about the influence of this parameter (M_polymer_) on the ability of cationic copolymer poly(AEMA-co-MAEMA-co-DMAEMA) to stabilize NPs (Table 3) [25].

Figure 5a presents absorption spectra of solutions containing NPs stabilized by cationic copolymers with various chain lengths.

It should be noted that the observed absorption spectra are approximately similar over a wide range of molar masses of stabilizing polymers (27.6–129.4 kDa). Only for the copolymer with M_polymer_ = 16.1 kDa, insignificant decrease in the intensity of maximum and its shift toward long-wave region were observed. This fact is apparently associated with the relatively low equilibrium rigidity of the used polymer, which in turn leads to a coil conformation of its macromolecules in the used M_polymer_ range. Under such conditions, the key parameter is the number of monomer units per AgNO_3_ molecule but not the M _polymer_. 

Hydrodynamic radii of individual stabilized nanoparticles and their aggregates obtained by dynamic light scattering (Table 3) similarly demonstrate no pronounced dependence on molar mass of stabilizing copolymer in the studied range of molar masses.

Sedimentation behaviour of solutions of NPs stabilized by cationic copolymers of various molar masses (AgNO_3_:monomer units = 4.88 ± 0.16) was studied by analytical ultracentrifugation (AUC). The experiments were carried out in water at 25 °C. Distributions of stabilized NPs by sedimentation coefficients are presented in Figure 5b and the corresponding data are given in Table 4. 

The resolved distributions turned out to be very wide: The found minimum and maximum values of sedimentation coefficients differ by more than one order of magnitude. However, judging by the values of optical density of solutions, in the course of the experiment at a rotor speed of 3000 to 5000 rpm, virtually all material moves from meniscus toward the bottom of sedimentation cell. Thus, it is reasonable to expect the presence of the main weight-average sedimentation coefficient that characterizes the studied solutions. It should be noted that the width of distributions by sedimentation coefficients satisfactorily correlates with that of distributions obtained by DLS (Figure 3, Table 3, Table 4). The revealed difference is mainly caused by specific features of the employed experimental approaches, mathematical models included in data processing and, of course, extreme complexity of the studied systems. The systems contain silver nanoparticles stabilized by copolymer, macromolecules that do not participate in stabilization and associates of nanoparticles. At the same time, dimensions of the present species (excluding associates) are almost similar, while their molar masses may differ considerably. Due to significant width of sedimentation distributions, it is not possible to quantitatively establish the influence of molar mass of stabilizing polymer on the final sizes of composite nanoparticles. The obtained data suggest, that cationic copolymers with molar masses ranging from 16.1 to 129.4 kDa allow stabilization of silver nanoparticles in aqueous solutions. When the protocol described above is adhered to, it is possible to prepare stabilized NPs with sedimentation coefficients (s) lying in the range from 800 to 1900 S (Table 4).

### 3.4. Analysis of Hydrodynamic Sizes of Stabilized NPs by AUC

Since the solutions of NPs stabilized by the copolymer with M_polymer_ = 27.6 kDa demonstrate the narrowest NP size distribution (Figure 5b), this system was selected for determination of concentration dependence of sedimentation coefficients of composite NPs. It was also used for solving a very important task: establishing partial specific volume of composite NPs. 

With this purpose, we studied the sedimentation velocity of the samples in H_2_O (ρ_0_ = 0.9971 g/cm^3^, η_0_ = 0.89 cP) and D_2_O (ρ_0_ = 1.1045 g/cm^3^, η_0_ = 1.099 cP) at 25 °C [34]. This approach was successfully used in the studies of various systems and consists in comparing intrinsic sedimentation coefficients of macromolecules in various solvents (provided that conformational parameters of the studied systems remain constant). In this case, determination of partial specific volume of composite NPs is complicated by their high polydispersity; therefore, estimation of this parameter required a number of experiments. The experiments were carried out over a wide range of polymer concentrations (Figure 6). This approach makes it possible to estimate partial specific volume of NPs stabilized by the polymer with M_polymer_ = 27.6 kDa and with the [AgNO_3_:monomer units] ratio equal to 5.00. The obtained value is $\bar{\upsilon }$ = (0.4 ± 0.2) cm^3^/g; however, the experimental error in this case is rather high. On the other hand, the found average value of $\bar{\upsilon }$ lies in the range (0.1–0.752), which is physically reasonable. The lowest limit corresponds to inverse density of silver and the highest limit corresponds to the partial specific volume that has been earlier determined for the initial polymer [25]. In our further estimations, we may assume that the found value of partial specific volume does not differ from that of composite NPs stabilized by other copolymers and at other ratios between AgNO_3_ and monomer units. The values of hydrodynamic radii (R_sf_) obtained with the use of these parameters are presented in Table 4.

The next part of our investigations included the so-called “differential sedimentation” approach. After sedimentation of composite nanoparticles at a rotor speed of 3 000 to 5 000 rpm, we observed sedimentation of the initial cationic polymer (M_polymer_ = 27.6 kDa) at 42 000 rpm [35,36]. The c(s) distributions obtained in the studies of solutions with various initial concentrations (indicated in the image) are presented in Figure 7a. We have also compared the results with the sedimentation coefficient values obtained in our earlier investigations (sedimentation of solutions of polymer with chlorine counterions) [25]. Figure 7b presents the intrinsic sedimentation coefficients with account for difference in the used solvents: The polymer with chlorine counterions was studied in 0.2 M NaCl and the current experiment was performed in H_2_O. Taking into account possible differences in the conformations of the polymers in solutions with different ionic strengths and the fact that the studied polymer was dialyzed in order to change counterions, the obtained experimental data might be considered reasonably correlated. The parameter characterizing shift of interference fringes (fringe, J) for the studied solutions was used to estimate concentration of “free” polymer. The following relationship was used: ∆n/∆c = Jλ/Kcl, where λ is the laser wavelength; K is the magnification coefficient; l is the optical path length [37]. Figure 7c presents comparison between the ∆n/∆c values obtained earlier (∆n/∆c = 0.145 ± 0.007) and the values corresponding to the distributions in Figure 7a ∆n/∆c = 0.063 ± 0.005. The latter value was estimated from the concentrations used during synthesis of composite NPs. Assuming that the ∆n/∆c value is constant, we can evaluate the fraction of polymer that does not take part in stabilizing silver nanoparticles; on the average, it is close to ~50% (Figure 7c).

Thus, the velocity sedimentation studies allowed to obtain independent data on the size of stabilized nanoparticles in solutions. In turn, this allowed to perform in-depth comparison with the results of DLS. The satisfactory intercorrelation of the results of both implemented techniques was demonstrated. The approach of “differential sedimentation” and the evaluation of sensitive optical characteristic of a solution (the refractive index increment) allowed to establish that approximately 50% of cationic copolymer present in the solution takes part in stabilization of Ag(0) nanoparticles. 

## 4. Conclusions

In the present work, we performed a comprehensive study of influence of various factors on stabilization of silver NPs with cationic copolymer poly(AEMA-*co*-MAEMA-*co*-DMAEMA) in aqueous medium.

It was demonstrated that varying the fraction of reducing agent (NaBH_4_) leads to linear dependence of NP concentration on the fraction of reducer in the range of molar ratios (NaBH_4_:AgNO_3_) from 0.23 to 0.99. Besides, in this range we observed considerable decrease in polydispersity of the obtained stabilized nanoparticles with increasing fraction of the reducer. Further increase in the reducer fraction leads to deterioration of spectral characteristics of the system. Note that all the solutions containing nanoparticles stabilized by copolymer show long-term stability of spectral parameters (for more than six months). The same is true for the case when the solution contains residual non-reduced silver Ag(+) (for molar ratios (NaBH_4_:AgNO_3_) < 1). This fact indicates that cationic copolymer poly(AEMA-*co*-MAEMA-*co*-DMAEMA) does not participate in reduction of Ag(+) and only stabilizes Ag(0) nanoparticles. 

It was demonstrated that the holding capacity of the copolymer changes linearly with increasing silver content up to the ratio AgNO_3_:monomer units = 5. In addition, in this concentration range we observed decrease in polydispersity of the stabilized nanoparticles with increasing silver fraction. With further increase in silver content, the dependence of holding capacity on molar ratio (AgNO_3_:monomer units) deviates from linearity and polydispersity of the final NPs increases.

The study of the influence of molar mass of the stabilizing polymer on its ability for stabilization showed that in the range of molar masses [27.6–129.4] kDa, virtually no dependence of the final NP concentration and their size distribution on M_polymer_ of copolymer is observed. When the polymer with a relatively low molar mass (M_polymer_ = 16.1 kDa) was used for stabilization, only an insignificant broadening of the absorption spectrum and a shift of the maximum towards the long-wave region was observed.

Sedimentation velocity studies allowed us to estimate sizes of stabilized nanoparticles in solutions; the results agree with the DLS data within the measurement error. It was also established that approximately 50% of cationic copolymer present in the solution takes part in stabilization of Ag(0) nanoparticles.

The obtained results are of great fundamental and practical importance for designing composites based on cationic copolymers containing NPs with a predicted set of physico-chemical characteristics. The further development of these studies is undoubtedly of great interest, since the copolymers allow a fine tuning of the physicochemical characteristics of the resulting systems over a wide range.

## Figures and Tables

**Figure 1 polymers-11-01647-f001:**
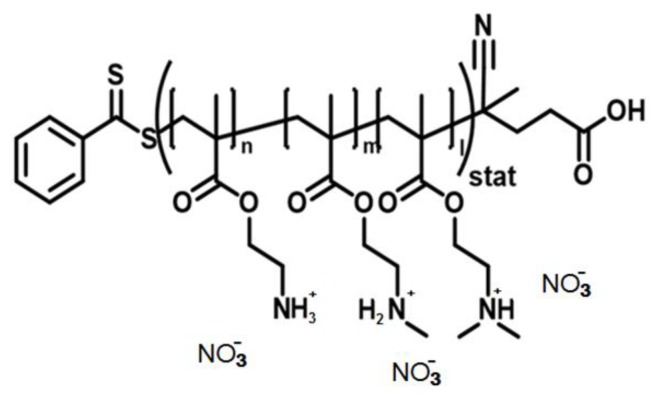
Schematic representation of the chemical structure of the cationic ternary copolymer poly(AEMA-*co*-MAEMA-*co*-DMAEMA); the average molar mass of a monomer unit (M_0_) is 143 Da.

**Figure 2 polymers-11-01647-f002:**
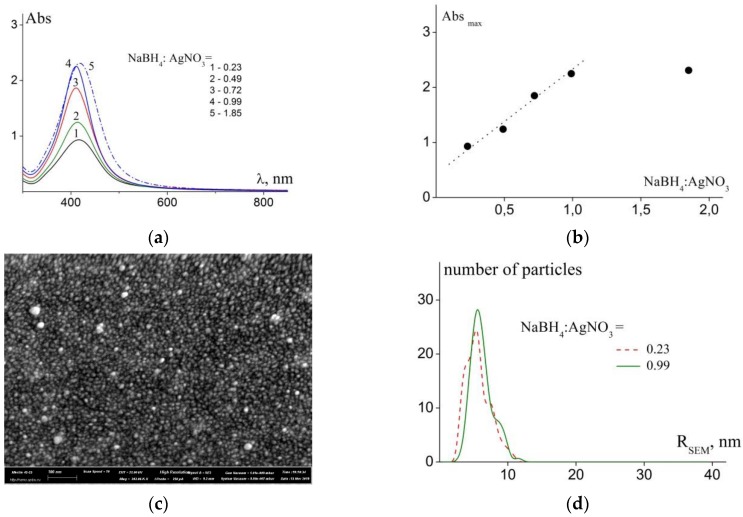
(**a**) Absorption spectra taken in three months after mixing at various molar ratios between NaBH_4_ and AgNO_3_ are indicated in the image; (**b**) Dependence of the intensity of absorption maximum on the fraction of reducer in equilibrium mixtures (c_mu_ = 0.49 mM); (**c**) Scanning electron microscopy (SEM) image of a polymer/NP sample (M_polymers_ = 27.6 kDa; NaBH_4_:AgNO_3_ = 0.99); (**d**) Size distributions of nanoparticle (NP) at various molar ratios between NaBH_4_ and AgNO_3_ (indicated in the image).

**Figure 3 polymers-11-01647-f003:**
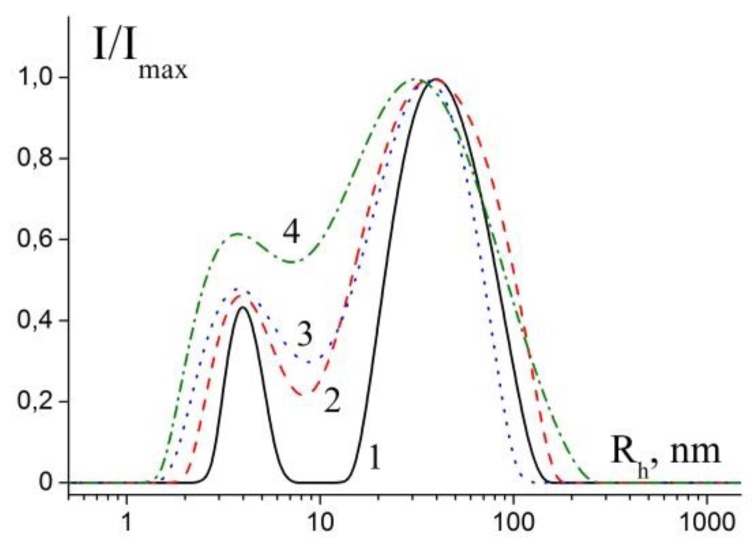
Normalized distribution function of hydrodynamic radii obtained from light scattering intensity for samples with various molar ratios between NaBH_4_:AgNO_3_: 0.99 (**1**), 0.72 (**2**), 0.49 (**3**), 0.23 (**4**). c_mu_ = 0.5 mM for all cases, scattering angle θ = 90°. M_polymers_ = 27.6 kDa.

**Figure 4 polymers-11-01647-f004:**
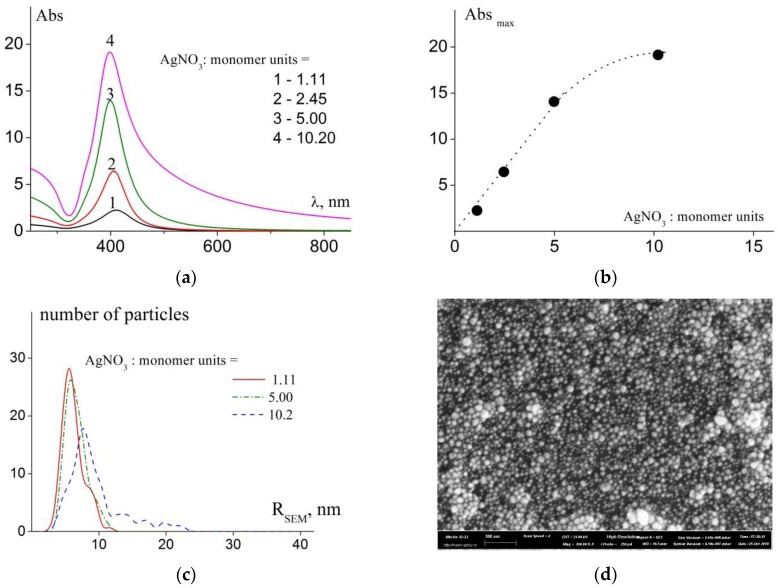
(**a**) Absorption spectra of mixtures with various ratios between AgNO_3_ and monomer units (indicated in the picture); (**b)** Dependence of absorption maximum intensity on the ratio between AgNO_3_ and monomer units (c_mu_ = 0.5 mM, molar ratio (NaBH_4_:AgNO_3_) = 0.96); (**c**) Size distributions for samples with various molar ratios between AgNO_3_ and monomer units (indicated in the picture); (**d**) SEM image of polymer/NP sample (M_polymer_ = 27.6 kDa, molar ratio (AgNO_3_:monomer units) = 5.00).

**Figure 5 polymers-11-01647-f005:**
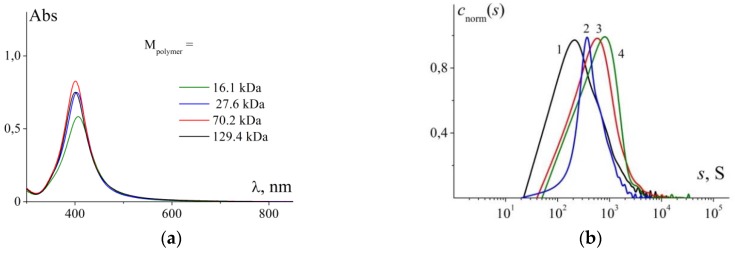
(**a**) Absorption spectra of solutions containing stabilizing copolymers with different molar masses (indicated in the image); c_mu_ = 0.035 mM for all solutions; (**b**) Normalized distributions c_norm_(s) of composite nanoparticles by sedimentation coefficients obtained for NPs stabilized by cationic copolymers with different M_polymer_, kDa: 129.4 (1), 27.6 (2), 70.2 (3) and 16.1 (4). The shift of sedimentation boundary was registered with the aid of absorption optical system at the monochromator wavelength of 404 nm. Copolymer concentration was varied from 0.3 to 0.6 mg/dL.

**Figure 6 polymers-11-01647-f006:**
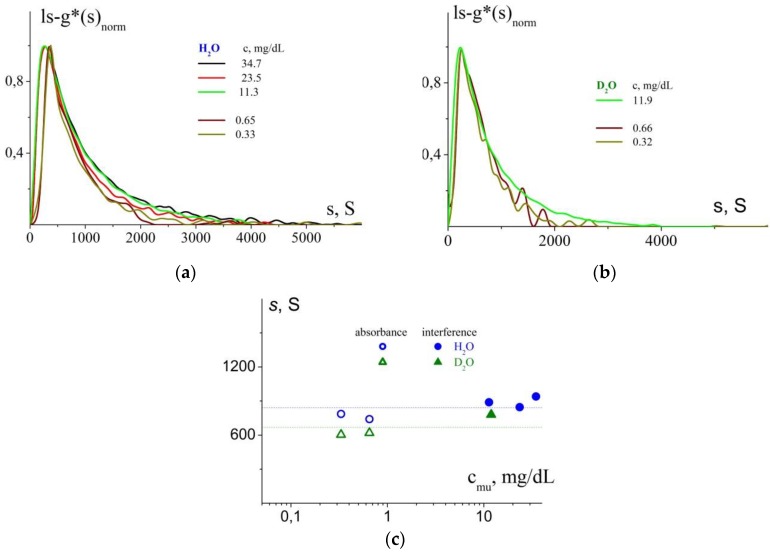
(**a**) Normalized distributions ls-g*(s)_norm_ of composite NPs (stabilized by the cationic copolymer with M_polymer_ = 27.6 kDa, at [AgNO_3_:monomer units] = 5.00) by sedimentation coefficients in H_2_O at various concentrations (indicated in the image); (**b**) Normalized distributions ls-g*(s)_norm_ of composite NPs (stabilized by the cationic copolymer with M_polymer_ = 27.6 kDa, at [AgNO_3_:monomer units] = 5.00) by sedimentation coefficients in D_2_O at various concentrations (indicated in the image); (**c**) Sedimentation coefficients at various concentrations in accordance with the distributions in Figure 6a,b. Open symbols: the concentration range registered by absorbance optics; filled symbols: the concentration range registered by interference optics. Dotted lines show averaged values of sedimentation coefficients obtained over the whole range of polymer concentrations.

**Figure 7 polymers-11-01647-f007:**
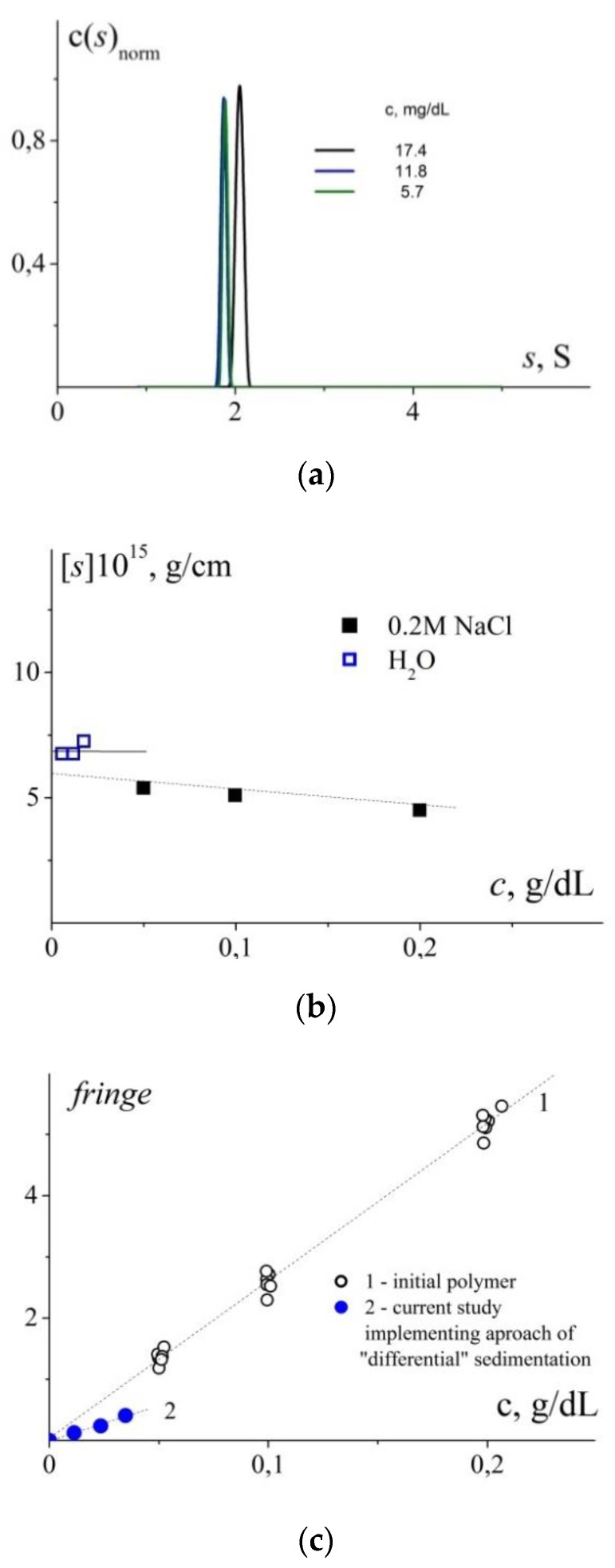
(**a**) Distributions of sedimentation coefficients for “free” copolymer (27.6 kDa) obtained using the “differential sedimentation” approach and resolved after sedimentation of composite NPs; (**b**) Comparison of intrinsic sedimentation coefficients found in Reference [25] (study of the initial polymer in 0.2 M NaCl) and the coefficients estimated from the distributions in Figure 7a; (**c**) Concentration dependences of the parameter characterizing shift of interference fringes obtained earlier (1) [25] and in the present work (2). The slope of dependence (1) corresponds to the average refractive index increment of the series of cationic copolymers in 0.2 M NaCl. Comparison between slopes of these dependences allows us to estimate the fraction of “free” polymer that does not participate in stabilization of silver nanoparticles.

**Table 1 polymers-11-01647-t001:** Influence of the NaBH_4_:AgNO_3_ molar ratio in the initial mixtures on hydrodynamic and spectral characteristics of the final stabilized NPs.

NaBH_4_: AgNO_3_	D_01_, 10^−^^7^ cm^2^/s	R_h1_, nm	D_02_, 10^−^^8^ cm^2^/s	R_h2_, nm	<R_h_>_c_ (PDI), nm	R_SEM_ (X), nm	λ_max_, nm	fwhm, nm
1.85	1.7	14.2	3.6	67	18 (2.3)	-	418	95
0.99	3.58	6.8	4.74	52	19 (2.4)	5.6 (2.8)	411	72
0.72	3.6	7.6	6.3	43	16 (2.5)	-	411	81
0.49	2.8	11.1	5.6	60	14 (2.7)	-	415	91
0.23	3.3	7.8	3.8	75	12 (2.4)	5.3 (3.4)	416	100

**Table 2 polymers-11-01647-t002:** Influence of molar ratio between AgNO_3_ and monomer units in the initial mixtures on hydrodynamic and spectral characteristics of the final stabilized NPs.

AgNO_3_:monomer units	D_01_, 10^−^^7^ cm^2^/s	R_h1_, nm	D_02_, 10^−^^8^ cm^2^/s	R_h2_, nm	<R_h_>_c_ (PDI), nm	R_SEM_ (X), nm	λ_max_, nm	fwhm, nm
1.11	3.58	6.8	4.74	52	19 (2.4)	5.6 (2.8)	411	72
2.45	5.60	4.4	11.2	22	12 (2.2)	-	405	57
5.00	4.9	5.3	9.0	27	7 (2.6)	5.8 (3.4)	402	57
10.2	1.43	17.1	-	-	10 (2.0)	7.7 (4.2)	398	91

**Table 3 polymers-11-01647-t003:** Influence of the molar mass of stabilizing polymer on the hydrodynamic and spectral characteristics of the final stabilized NPs.

M_polymer_, kDa	D_01_, 10^−^^7^ cm^2^/s	R_h1_, nm	D_02_, 10^−^^8^ cm^2^/s	R_h2_, nm	R_h_ ^*^, nm	λ_max_, nm	fwhm, nm
16.1	4.9	5.0	10.0	24	2.7	407	61
27.6	4.9	5.3	9.0	27	3.5	402	57
70.2	5.4	4.5	10.6	23	5.9	401	53
129.4	2.4	10.1	5.3	46	8.7	400	55

^*^ [25].

**Table 4 polymers-11-01647-t004:** Influence of molar mass of stabilizing polymer on distribution of sedimentation coefficients in the final mixtures (NPs stabilized by the copolymer).

M_polymer_, kDa	s, S	R_sf._, nm	<R_h_>_c_ (PDI), nm	R_h1_/R_h2_ [DLS], nm
16.1	1400	19 (12–29) *	17 (2.7)	5.0/24
27.6	840	15 (9–22) *	7 (2.6)	5.3/27
70.2	1900	22 (14–34) *	11 (2.1)	4.5/23
129.4	1100	17 (11–26) *	13 (2.1)	10.1/26

* the range of values is determined by the found measurement error of the $\bar{\upsilon }$ values.

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
