# Peer review of "Stabilization of Silver Nanoparticles by Cationic Aminoethyl Methacrylate Copolymers in Aqueous Media—Effects of Component Ratios and Molar Masses of Copolymers"

_polymers, 2019, doi:10.3390/polym11101647_

Round 1
Reviewer 1 Report
The research problem is well defined. The characterization techniques are well explained The discussion can be with more detail so as to make more clear of the results.
The research is so interesting, specially the techniques used in order to obtain the effect of molar mass in the stabilization of nanoparticles and the sedimentation studies.
It is necessary to present a experimental design and statistical analysis in order to obtain more strong results and conclusions.
Author Response
Dear Referee,
We are very grateful for attention to the manuscript and valuable comments regarding its content. We have thoroughly analyzed the raised questions and prepared step-by-step answers together with manuscript corrections (marked with red font), where it was necessary. We also prepared a file "Supplemental Material", which included examples of initial experimental data and intermediate stages of their analysis to illustrate the quality of the studies.
Referee 1.
Comments and Suggestions for Authors
The research problem is well defined. The characterization techniques are well explained The discussion can be with more detail so as to make more clear of the results.
The research is so interesting, specially the techniques used in order to obtain the effect of molar mass in the stabilization of nanoparticles and the sedimentation studies.
It is necessary to present a experimental design and statistical analysis in order to obtain more strong results and conclusions.
A file “Supplementary Material” (SM) was prepared and uploaded with the corrected manuscript. It contains raw experimental data and the results with their direct processing for clear illustration of its quality. The corresponding corrections are also made within the manuscript itself (marked with red font).
Reviewer 2 Report
This manuscript described a work where it was studied the ability of aminoethyl cationic copolymers in the stabilization of silver nanoparticles in aqueous media. Furthermore, it is evaluated the influence of diverse factors, namely the ratio of reducing agent – concentration of AgNO3, monomer units – concentration of AgNO3, and the molecular weight of the copolymers on the stabilization of silver NPs using cationic copolymer poly(AEMA-co-MAEMA-co-DMAEMA).
The characterization is well done; however, the SEM images are unreadable for an accurate determination of the size and the polydispersity index along with the DLS data, is missing. Since the size and polydispersity index are critical parameters in this work, a TEM analysis is probably required due to the small size presented by the AgNPs in order to have a valid confirmation of these parameters.
Other concern is regarding the novelty of the work. This is an incremental work since literature, namely two works developed by these authors, already describe the use of the same copolymer for the same purpose (stabilization of aqueous solutions of silver nanoparticles). The work described in this paper only study the influence of several parameters of an already published procedure. As original work, it misses an explanation for the observed phenomena instead of a simple description of the events. I also suggest a critical comparison of the observations with other similar systems described in the literature.
In order to improve the document, other points in the text need to be changed and/or clarified.
Abstract: The authors claim that the particles are stable for over six months, but analysis after the preparation of AgNPs and a continuous study of the stability of the particles overtime is mandatory to understand this behavior.
Page 2, Lines 74-75: In my view, since the primary objective of this work is the study of the polydispersity index the authors need to put in the tables the value obtained in DLS for the PdI. Furthermore, a TEM analysis of the samples will allow the calculation of average size and respective standard deviation. These two parameters will permit to have a more accurate evaluation of the “real” polydispersity index of the samples instead only the value of the fwhm obtained in the UV-vis analysis.
Page 4, Lines 148-153: The NPs after the synthesis where washed to remove the initial unreacted reagents? Since any washing step is described the presence of these unreacted moieties could not influence the results obtained?
Page 5, Lines 175-176: The authors claimed that “Figure 2a presents the absorption spectra taken after establishing equilibrium in the system”. How they now that the equilibrium is reached after six days? Since the NaBH4: AgNO3 molar ratio was modified, probably the time for equilibrium for the samples is distinct due to the different amount of reducing agent used. A UV-vis analysis overtime is necessary.
Figure 2a: Why the authors put an absorption spectra of tree months since in the test refer six days and stability of 6 months?
Figure 2c and Figure 4d: The unit represented in the scale bar of the SEM micrographs is HM. Does this correspond to nanometers (nm)?
Figure 2d and Figure 4c: These graphs are not histograms.
Table 1: Why the values of RSEM (x) are missed for some samples?
Page 6, Lines 209-211: I do not agree with this statement. When SEM analysis is performed, it is possible to see also the NPs stabilized by the polymer chains. However, for a better understanding of the system, a TEM analysis is recommended.
Page 7, Line 227: The authors could explain in the discussion the importance of measure the holding capacity for these materials.
Page 7, Line 230: For example, where the authors analyze the samples two months after the synthesis. The authors need to use well-defined time throughout the document in order to allow a better and reliable comparison of all parameters studied.
Page 8, Lines 262-266: The DLS analysis shows an increase in the size of the particles not higher polydispersity. The introduction by the authors of the PdI values obtained in DLS analysis could help in the validation of these statements.
Page 9, Lines 286-288: How the authors explained that in the UV-vis spectra in Figure 5a the intensity of the band correspondent of SPR of AgNPs change in the respective intensity since the concentration of AgNO3 and respective reducing reagent is the same?
Page 10, Line 324: For a reliable study of the NPs stability, that the authors can use to compare with the results obtained in this point (3.4), they can analyze the radii or diameter of the particles over time, for example using DLS, analyzing the same colloidal suspension at different time points (e.g., 0h, 1h, 24h, 1 week, 2 weeks, 1 month …).
Author Response
Dear Referee,
We are very grateful for attention to the manuscript and valuable comments regarding its content. We have thoroughly analyzed the raised questions and prepared step-by-step answers together with manuscript corrections (marked with red font), where it was necessary. We also prepared a file "Supplemental Material", which included examples of initial experimental data and intermediate stages of their analysis to illustrate the quality of the studies.
Referee 2.
Comments and Suggestions for Authors
This manuscript described a work where it was studied the ability of aminoethyl cationic copolymers in the stabilization of silver nanoparticles in aqueous media. Furthermore, it is evaluated the influence of diverse factors, namely the ratio of reducing agent – concentration of AgNO3, monomer units – concentration of AgNO3, and the molecular weight of the copolymers on the stabilization of silver NPs using cationic copolymer poly(AEMA-co-MAEMA-co-DMAEMA).
The characterization is well done; however, the SEM images are unreadable for an accurate determination of the size and the polydispersity index along with the DLS data, is missing. Since the size and polydispersity index are critical parameters in this work, a TEM analysis is probably required due to the small size presented by the AgNPs in order to have a valid confirmation of these parameters.
Within manuscript body, SEM images have been replaced with once of higher resolution (Figures 2c and 4d). Also additional SEM images were introduced to “Supplementary material” file. It can be easily seen, that the dried samples represent clusters of nanoparticles (due to the fact that the samples were prepared by drying droplets of solutions on silicon wafers at 45 °C and further the study was carried out on the edge parts of dried droplets). Such a high concentration of nanoparticles leads to the fact that partial overlapping of nanoparticles is observed at the images, which obviously complicates the standard processing of the obtained image. Histograms (figures 2d and 4c, figures 5 SM) were obtained by finding the symmetric radii from over 100 images of individual NPs on SEM images in each case (using free and open source software Gwyddion) and directly counting the number of particles falling in different intervals (with an increment of 1 nm).
The mean values of hydrodynamic radius <Rh>c and polydispersity index (PDI) were obtained by cumulant analysis. The corresponding data was included in tables 1, 2 and 4. Also the cumulant analysis approach was introduced in section 2.1. Methods (marked with red font, Lines 92-100).
A comparison of the obtained <Rh>c values with the velocity sedimentation data demonstrates their satisfactory intercorrelation.
However, we want to emphasise, that the system under study is quite complicated. It is a solution containing three main components: copolymer macromolecules, individual nanoparticles stabilized by copolymer molecules and aggregates comprising a different number of nanoparticles stabilized by macromolecules of copolymers.
UV-vis and SEM studies have allowed us to conclude that silver NPs have a fairly narrow size distribution, and the dispersion of particles found in solution is mostly associated with the dispersion of the third solution component — aggregates.
Undoubtedly, the reviewer is absolutely right that the information obtained by the TEM method could be useful for a more detailed study of nanoparticles. However, at this stage, the SEM method has sufficient resolution to determine the size of the NPs and to confirm the fact that the synthesized NPs have a fairly narrow size distribution.
It should be emphasized that the main focus of the presented manuscript is directed to the study of synthesized systems in solution. Hydrodynamic methods (DLS and AUC) demonstrate a fairly wide distribution of the determined characteristics, because those reflect the entire spectrum of objects present in the solution, and not just individual nanoparticles.
One of the most important features of the DLS method is that it is very sensitive to the presence of large particles in the system. For particles with optically equivalent characteristics, the contribution of each component to the total light scattering intensity is proportional to the volume, and for particles with a shape close to their spherical radius in the cube I ~ V ~ R3. The paper analyzes the DLS results by the regularization method, It allows one to obtain the distribution of the scattered light intensity over relaxation times / hydrodynamic radii. The exact calculation of the fraction of each of the components in our case is a complex and probably not solvable task in the framework of this work. Evaluation shows that the presence of large particles in the solution are quite small, less than 1%. Taking this fact into account, It becomes clear that the calculation of PDI by regularization from DLS data is significantly difficult task.
Estimated PDI values by the cumulant method were added to the manuscript. This method has its own certain drawbacks and it is not the most accurate one. Although It is easy to implement, which makes it widely distributed. The presence of these data in the work will improve the perception of the results.
Other concern is regarding the novelty of the work. This is an incremental work since literature, namely two works developed by these authors, already describe the use of the same copolymer for the same purpose (stabilization of aqueous solutions of silver nanoparticles). The work described in this paper only study the influence of several parameters of an already published procedure. As original work, it misses an explanation for the observed phenomena instead of a simple description of the events. I also suggest a critical comparison of the observations with other similar systems described in the literature.
We would like to note that so far we have published only one work devoted to the stabilization of silver NPs by a cationic copolymer of this composition [27]. In this work, essentially only the possibility of long-term stabilization of silver nanoparticles by a copolymer of this composition is presented. The work now being presented is a large-scale systematic study of the influence of various factors on the final distribution of nanoparticles in the system, which most modern papers are deprived of [20, 21, etc.]. Only a few authors undertake the work of studies of such type [18, 1. Si, S .; Kotal, A .; Mandal, T.K .; Giri, S .; Nakamura, H .; Kohara, T. Size-Controlled Synthesis of Magnetite Nanoparticles in the Presence of Polyelectrolytes. Chem. Mater. 2004, 16, 3489–3496; 2.Kazim, S .; Jäger, A .; Steinhart, M .; Pfleger, J .; Vohlídal, J .; Bondarev, D .; Štěpánek, P. Morphology and Kinetics of Aggregation of Silver Nanoparticles Induced with Regioregular Cationic Polythiophene. Langmuir 2016, 32, 2-11]. In this regard, at the moment it is not possible to conduct a comprehensive comparison of our results with data from other authors. However, as shown in this paper, such studies lead to a significant improvement in the spectral properties of the resulting nanoparticles. In our opinion, such a planned approach is extremely useful in developing strategies for the synthesis of nanoparticles using polymers as stabilizers.
As for our explanation of the observed phenomena, we made appropriate additions at the each paragraph (3.1 - 3.4), summarizing the results of the part of the study (marked with red font).
In order to improve the document, other points in the text need to be changed and/or clarified.
Abstract: The authors claim that the particles are stable for over six months, but analysis after the preparation of AgNPs and a continuous study of the stability of the particles overtime is mandatory to understand this behavior.
Absorption spectra measured since start of stabilization and at different ratios of NaBH4: AgNO3 and AgNO3: monomer units (figures 1 SM, 2 SM) were introduced to SM file.
It should be noted that the spectra at low NaBH4: AgNO3 ratios (0.23, 0.49) show a small (about 30%) increase during the year after mixing of the components. At NaBH4: AgNO3 ratios of 0.72 and 0.99, no such changes are observed. A further increase in the proportion of the reducing agent leads to a small decrease in the peak value of the plasmon resonance.
To study holding capacity, we selected the ratio NaBH4: AgNO3 = 0.99.
The data presented at Figure 4 SM demonstrate long-term stability for all AgNO3: monomer units ratios.
Molar mass influence on the results of stabilization has been studied at the following component ratios (NaBH4: AgNO3 ) = 0.96 and (AgNO3: monomer) = 4.88, i. e. at those ratios which satisfied the condition of long term spectra stability while studying the polymer with Mpolymer = 27.6 kD.
Figures 1, 3 (SM) show the time dependence of the peak maxima of plasmon resonance.
In studying the effect of Mpolymer, the time dependences were not measured in detail; only control measurements were carried out, which fully confirmed the described above observations.
Page 2, Lines 74-75: In my view, since the primary objective of this work is the study of the polydispersity index the authors need to put in the tables the value obtained in DLS for the PdI. Furthermore, a TEM analysis of the samples will allow the calculation of average size and respective standard deviation. These two parameters will permit to have a more accurate evaluation of the “real” polydispersity index of the samples instead only the value of the fwhm obtained in the UV-vis analysis.
First of all, We would like to emphasize that the dispersity of the already stabilized NPs is discussed here. To elaborate on the determined parameter We added the mean values of hydrodynamic radius <Rh>c and polydispersity index (PDI) obtained by cumulant analysis to tables 1, 2 and 4 with further analysis of obtained results.
Page 4, Lines 148-153: The NPs after the synthesis where washed to remove the initial unreacted reagents? Since any washing step is described the presence of these unreacted moieties could not influence the results obtained?
This is absolutely correct: after the synthesis, no additional procedures had been performed with the systems. The objective of this study was not only to obtain information on the total spectral characteristics and sizes of stabilized NPs by varying this parameter, but also to confirm experimentally that the studied copolymer is not a reducer of silver ions not only in a copolymer + AgNO3 mixture in an aqueous medium, but also in the presence of mixtures of a variable amount of already stabilized nanoparticles. The fact that no significant changes are observed over a sufficiently long period of time in the absorption spectra confirms the above assumption, which is of great practical importance. Washing the system after restoration would not give us the opportunity to get the above confirmation.
An attempt to somehow get rid of free (not participating in the stabilization of polymer molecules) will lead to a change in the ratio (AgNO3: monomer units) and, as a result, this may lead to the processes described in the polymer holding capacity part. Although of course, this assumption requires additional verification
Page 5, Lines 175-176: The authors claimed that “Figure 2a presents the absorption spectra taken after establishing equilibrium in the system”. How they now that the equilibrium is reached after six days? Since the NaBH4: AgNO3 molar ratio was modified, probably the time for equilibrium for the samples is distinct due to the different amount of reducing agent used. A UV-vis analysis overtime is necessary.
The absorption spectra shown in Figs. 1 SM and 2 SM together with the time dependences of the maxima of the plasmon resonance peak allowed us to conclude that the establishment of equilibrium is practically independent of the NaBH4: AgNO3 ratio. The peak value of plasmon resonance, which is established after 6 days, changes further only by no more than a few percent during the month, and then the system remains practically stable for a long time. Based on this observation, we allowed ourselves to conduct research using all the other methods used (more labor and time consuming) not at once, but through some time intervals and compare the results with each other.
Figure 2a: Why the authors put an absorption spectra of three months since in the test refer six days and stability of 6 months?
As it was stated above, the most graphic diagrams are given for all variations of the ratios that we studied. We emphasize that equilibrium occurs in 6 days and up to 6 months remains unchanged, which can be seen in the graphs 1 SM - 4 SM.
Figure 2c and Figure 4d: The unit represented in the scale bar of the SEM micrographs is HM. Does this correspond to nanometers (nm)?
The corresponding changes have been made (Figure 2c and Figure 4d).
Figure 2d and Figure 4c: These graphs are not histograms.
The corresponding changes have been made (Figure 2d and Figure 4c).
The methods part was revised accordingly.
Table 1: Why the values of RSEM (x) are missed for some samples?
Where RSEM (x) values are not specified, SEM images were not obtained. Close results at the boundary points of the ranges when varying various parameters (values (NaBH4: AgNO3) = 0.99, 0.23 - Table 1 and values (AgNO3: monomer units) = 1.11, 5.00 - Table 2) justify the absence of the need for intermediate studies.
Page 6, Lines 209-211: I do not agree with this statement. When SEM analysis is performed, it is possible to see also the NPs stabilized by the polymer chains. However, for a better understanding of the system, a TEM analysis is recommended.
The corresponding clarification and changes were made (lines 216 - 218).
Page 7, Line 227: The authors could explain in the discussion the importance of measure the holding capacity for these materials.
The corresponding clarification and changes were made (lines 297 - 299).
Page 7, Line 230: For example, where the authors analyze the samples two months after the synthesis. The authors need to use well-defined time throughout the document in order to allow a better and reliable comparison of all parameters studied.
We agree that a comparison of experimental results obtained at strictly fixed points in time throughout the paper would provide a better and more reliable comparison of all studied parameters. Although, planning such a complex and time-consuming study is not an easy task and is associated with certain difficulties (not always dependent on the experimenter). When comparing the results obtained by different methods, we tried to compare the results obtained at close time points, indicating those in the captions under the graphs. In this assumption, we relied on knowledge of the temporal stability of the absorption spectra (figures 1 SM - 4 SM).
The corresponding clarification and changes were made (line 253).
Page 8, Lines 262-266: The DLS analysis shows an increase in the size of the particles not higher polydispersity. The introduction by the authors of the PDI values obtained in DLS analysis could help in the validation of these statements.
The mean values of hydrodynamic radius <Rh>c and polydispersity index (PDI) obtained by cumulant analysis were introduced to Tables 1, 2 and 4 with further analysis of obtained results.
Page 9, Lines 286-288: How the authors explained that in the UV-vis spectra in Figure 5a the intensity of the band correspondent of SPR of AgNPs change in the respective intensity since the concentration of AgNO3 and respective reducing reagent is the same?
In our opinion, for example, this may be due to insufficient polymer chains length for stabilization. Unfortunately, we do not have enough data for a clear answer. All that we can conclude on the basis of the results obtained, that a pronounced dependence is not observed upon variation of the molecular weight range of the used polymer in the studied (eightfold) range. However, for a sample with a lower Mpolymer, a slightly wider size distribution of the objects present in the mixture was found.
Page 10, Line 324: For a reliable study of the NPs stability, that the authors can use to compare with the results obtained in this point (3.4), they can analyze the radii or diameter of the particles over time, for example using DLS, analyzing the same colloidal suspension at different time points (e.g., 0h, 1h, 24h, 1 week, 2 weeks, 1 month …).
In our study, we primarily focused on systems in which the silver reduction reaction has already completed. Undoubtedly, the study of the dynamics of the formation of silver nanoparticles in the presence of polymers is an important task, and indeed, DLS can be used to solve it. It should be taken into account that the characteristics of the initial copolymers were determined by us earlier and presented in [25], and in the process of nanoparticle synthesis the mixture is under condition of continuous stirring. The changes occurring at this time were recorded only using the rather fast UV method.
Based on the temporal stability of the absorption spectra, we nevertheless tried to compare the results of studies with various methods at close time points from the introduction of the reducing agent.
Moreover section 3.4 is devoted to the study of stable NPs by analytical ultracentrifugation. Fundamentally, the features of this method do not suggest its use for the analysis of systems in non-stationary modes. During centrifugation, the local concentration of the polymer in the solution changes, which may affect the final characteristics of the resulting objects.
Reviewer 3 Report
The work is devoted to the study of the influence of the polymer matrix, reagents and their ratio on the ability to stabilize silver nanoparticles and size distribution. The work is a continuation and more detailed study of [27].
The field of preparation and research of polymer nanocomposites filled with silver particles is rather well understood, so its quality complement - not an easy task.
As the polymer stabilizing matrix, the authors chose an exotic cationic copolymer of poly((2-aminoethyl)methacrylate-co-N-methyl(2-aminoethyl)methacrylate-co-N,N-dimethyl(2-aminoethyl)meth-acrylate), the synthesis of which is a separate task [25]. In this regard, the question arises of how reasonable it is to use a similar matrix to obtain a silver polymer nanocomposite. Indeed, there are much simpler, widespread matrices with good stabilizing ability [14,16,18-21, for example].
Silver in nanoscale form is not the most difficult metal for stabilization. Most often, researchers point to the antibacterial properties of silver nanoparticles in the prospects for using the obtained composites. For cancer treatment, the particles must have good unimodality to selectively penetrate the cell membranes of cancer cells. For struggle, for example, with S.aureus and E.coli, the requirement for unimodality is much less, but this often involves the use of available simple polymers and gels as stabilizing matrices.
The authors repeatedly note the absence of aggregation processes and changes in the absorption spectra of samples for more than 6 months (lines 25, 70, 334). Then, for example, why not cite the comparison results obtained immediately after the introduction of the reducing agent and after 6 months? Line 175 refers to the spectra obtained after 6 days of introducing the reducing agent, and the signature to Figure 2 a) (line 181) refers to 3 months, which of this is true? And why exactly 3 months? Could you explain how the histogram Fig.2d) and Fig 4c) was compiled? Lines 204–206 indicate good agreement between the results of SEM-studies, histogram 2d) and data from the UV-spectra. However, at the indicated scale (100 nm) in Fig. 2 c) the average particle size is 2 times larger (~10 nm) than that described by the authors. The same question arises in Figure 4d); in addition, the dispersion of particles there is visually much larger than in the histogram 4c)
In general, the article is written in an accessible, logical way. The experimental methods are described in detail. The experiments are thought out, and the results are presented in a structured and understandable way. The study was conducted in depth and in detail. The authors used interesting methods to characterize the described nanocomposites.
From the point of view of the fundamental science of nanocomposites, the article expands understanding of the methods for producing materials with set properties. It’s interesting to read.
Author Response
Dear Referee,
We are very grateful for attention to the manuscript and valuable comments regarding its content. We have thoroughly analyzed the raised questions and prepared step-by-step answers together with manuscript corrections (marked with red font), where it was necessary. We also prepared a file "Supplemental Material", which included examples of initial experimental data and intermediate stages of their analysis to illustrate the quality of the studies.
Referee 3.
Comments and Suggestions for Authors
The work is devoted to the study of the influence of the polymer matrix, reagents and their ratio on the ability to stabilize silver nanoparticles and size distribution. The work is a continuation and more detailed study of [27].
The field of preparation and research of polymer nanocomposites filled with silver particles is rather well understood, so its quality complement - not an easy task.
As the polymer stabilizing matrix, the authors chose an exotic cationic copolymer of poly((2-aminoethyl)methacrylate-co-N-methyl(2-aminoethyl)methacrylate-co-N,N-dimethyl(2-aminoethyl)meth-acrylate), the synthesis of which is a separate task [25]. In this regard, the question arises of how reasonable it is to use a similar matrix to obtain a silver polymer nanocomposite. Indeed, there are much simpler, widespread matrices with good stabilizing ability [14, 16, 18-21, for example].
Of course, at the moment, many polymer systems, including cationic ones, with good stabilizing ability are already known. However, this scientific area of research cannot be considered fully completed. A large number of studies are now devoted to copolymers containing monomeric units with different functional loads (both statistically and block or gradient distributed over the chain).
This fact is of great practical importance, since cationic polymers are actively used as carriers in the transfection of DNA into cells, as well as in the treatment of various diseases (1. Ramamoorth M., Narvekar A. // J. Clin. Diagn. Res. 2015. V 9. No. 1. P. GE01; 2. Kabanov AV, Astafyeva IV, Chikindas ML, Rosenblat GF, Kiselev VI, Severin ES, Kabanov VA // Biopolymers. 1991. V. 31. P. 1437; 3. Shi B ., Zheng M., Tao W., Chung R., Jin D., Ghaffari D., Farokhzad OC // Biomacromolecules. 2017. V. 18. P. 2231; 4. De Cock LJ, De Koker S., De Geest BG, Grooten J., Vervaet C., Remon JP, Sukhorukov GB, Antipina MN // Angew. Chem. Int. Ed. 2010. V. 49. P. 6954. etc.).
The creation of complexes of cationic polymers with NPs can significantly expand the field of practical application of such systems.
The present work at this stage does not contribute much to expanding the list of cationic polymers capable of long-term stabilization of nanoparticles. On the other hand, it is a step towards understanding the role of individual copolymer groups in the stabilization process and the influence of the copolymer structure on this role. The presence in our group of homologous series of copolymer samples poly(AEMA-co-MAEMA) and poly(AEMA-co-DMAEMA) as well as polymer poly(AEMA) and stabilization experiments with them are now giving us hope for clarification of this issue in the following planned study.
The corresponding clarification was made (lines 459 - 461).
Silver in nanoscale form is not the most difficult metal for stabilization. Most often, researchers point to the antibacterial properties of silver nanoparticles in the prospects for using the obtained composites. For cancer treatment, the particles must have good unimodality to selectively penetrate the cell membranes of cancer cells. For struggle, for example, with S.aureus and E.coli, the requirement for unimodality is much less, but this often involves the use of available simple polymers and gels as stabilizing matrices.
The authors repeatedly note the absence of aggregation processes and changes in the absorption spectra of samples for more than 6 months (lines 25, 70, 334). Then, for example, why not cite the comparison results obtained immediately after the introduction of the reducing agent and after 6 months?
The corrections have been made. The above mentioned absorption spectra have been introduced to SM file.
Line 175 refers to the spectra obtained after 6 days of introducing the reducing agent, and the signature to Figure 2 a) (line 181) refers to 3 months, which of this is true? And why exactly 3 months?
In this particular place, the phrase was not entirely correct. “In six days after introduction of the reducing agent” is a specification of the moment of equilibrium establishment, and not of the moment when the indicated spectra were obtained. We have made the necessary corrections.
Based on the fact that the studied systems remain stable for a long period of time (SM, figures 2 SM, 4 SM), we did not always compare the results of studies with different methods obtained simultaneously, but we tried to compare the results obtained at close time points, indicating those in signatures under the graphs.
Could you explain how the histogram Fig.2d) and Fig 4c) was compiled? Lines 204–206 indicate good agreement between the results of SEM-studies, histogram 2d) and data from the UV-spectra. However, at the indicated scale (100 nm) in Fig. 2 c) the average particle size is 2 times larger (~10 nm) than that described by the authors. The same question arises in Figure 4d); in addition, the dispersion of particles there is visually much larger than in the histogram 4c)
As can be seen in the SEM images (figures 2c and 4d) presented in the article and in SM file, the dried samples represent clusters of nanoparticles (due to the fact that the samples were prepared by drying droplets of solutions on silicon wafers at 45 °C and further the study was carried out on the edge parts of dried droplets). Such a high concentration of nanoparticles leads to the fact that partial overlapping of nanoparticles is observed at the images, which obviously complicates the standard processing of the obtained image. Distributions (figures 2d and 4c) were obtained by finding the symmetric radii from over 100 images of individual NPs on SEM images in each case (using free and open source software Gwyddion) and directly counting the number of particles falling in different intervals (with an increment of 1 nm).
We added additional SEM raw data to SM file.
In general, the article is written in an accessible, logical way. The experimental methods are described in detail. The experiments are thought out, and the results are presented in a structured and understandable way. The study was conducted in depth and in detail. The authors used interesting methods to characterize the described nanocomposites.
From the point of view of the fundamental science of nanocomposites, the article expands understanding of the methods for producing materials with set properties. It’s interesting to read.
Round 2
Reviewer 2 Report
The authors made all the corrections suggested and at the same time, added new data and supplementary information for a better perception of the results of the article, increasing the value of the work.